# Identification of risk factors and mosquito vectors associated with dengue virus infection in American Samoa, 2017

Tyler M. Sharp[1,2]*, A. John Tufa[3,4], Caitlin J. Cotter[1,2,5], Matthew J. Lozier[1,2], Gilberto A. Santiago[1], Stephanie S. Johnson[1,6], Mary Mataia'a[5], Stephen H. Waterman[1,2], Jorge L. Muñoz-Jordán[1], Gabriela Paz-Bailey[1], Ryan R. Hemme[1], Mark A. Schmaedick[7], Scott Anesi[4]

1 Dengue Branch, Centers for Disease Control and Prevention, San Juan, Puerto Rico, 2 United States Public Health Service, Silver Springs, Maryland, United States of America, 3 Pacific Island Health Officers' Association, Honolulu, Hawaii, United States of America, 4 American Samoa Department of Health, Pago Pago, American Samoa, 5 Epidemic Intelligence Service, Centers for Disease Control and Prevention, Atlanta, Georgia, United States of America, 6 Applied Epidemiology Fellowship, Council of State and Territorial Epidemiologists, Atlanta, Georgia, United States of America, 7 American Samoa Community College, Pago Pago, American Samoa

* tsharp@cdc.gov

**Data Availability Statement:** Data will be made available online in the form of a supplemental file.

## Abstract

### Introduction

The first outbreak of dengue in American Samoa was reported in 1911. Sporadic outbreaks have been reported since, as were outbreaks of other pathogens transmitted by *Aedes* species mosquitoes including Ross River, chikungunya, and Zika viruses. During an outbreak of dengue virus-type 2 (DENV-2) in 2016–2018, we conducted household-based cluster investigations to identify population-specific risk factors associated with infection and performed entomologic surveillance to determine the relative abundance of *Ae. aegypti* and *Ae. polynesiensis*.

### Methods and findings

We contacted dengue patients who had tested positive for DENV infection and offered them as well as their household members participation in household-based cluster investigations. For those that accepted participation, we also offered participation to residents of households within a 50-meter radius of each case-patient's home. Questionnaires were administered and serum specimens collected for testing by RT-PCR and anti-DENV IgM ELISA. Adult female mosquitoes were aspirated from inside and outside participating households and tested by RT-PCR. We analyzed characteristics associated with DENV infection in bivariate analyses. A total of 226 participants was enrolled from 91 households in 20 clusters. Median age of participants was 34 years (range: <1–94), and 56.2% were female. In total, 7 (3.2%) participants had evidence of DENV infection by IgM ELISA (n = 5) or RT-PCR (n = 2). Factors significantly associated with DENV infection were reporting a febrile illness in the past three months (prevalence ratio: 7.5 [95% confidence interval: 1.9–29.8]) and having a household septic tank (Fisher's Exact Test, p = 0.004). Of 93 *Ae. aegypti* and 90 *Ae.*

**Funding:** The authors received no specific funding for this work.

**Competing interests:** The authors have declared that no competing interests exist.

*polynesiensis* females collected, 90% of *Ae. aegypti* were collected inside homes whereas 83% of *Ae. polynesiensis* were collected outside homes. DENV nucleic acid was not detected in any mosquito pools. Sequencing of the DENV-2 from patient specimens identified the Cosmopolitan genotype of DENV-2 and was most closely related to virus detected in the Solomon Islands during 2016.

## Conclusions

This investigation demonstrated that dengue is a continuing risk in American Samoa. Increased frequency of infection among residents with a septic tank suggests a need to investigate whether septic tanks serve as larval habitats for mosquito vectors of DENV in American Samoa. Future efforts should also evaluate the role of *Ae. polynesiensis* in DENV transmission in the wild.

## Introduction

Dengue is the most common mosquito-borne viral disease worldwide and a leading cause of morbidity throughout the tropics, where an estimated 50 million symptomatic infections occur each year [1–4]. Following infection with any of the four mosquito-transmitted dengue virus-types (DENV-1–4), illness can range from a mild, nonspecific acute febrile illness (AFI) to severe dengue characterized by increased intravascular permeability leading to shock, and hemorrhage [1].

American Samoa is a territory of the United States (U.S.) located in the South Pacific Ocean. Multiple reports of dengue outbreaks were made from American Samoa during 1911–1945 [5], and DENV was first detected by the U.S. Navy in 1972 [6]. Soon after, an epidemic of Ross River virus occurred throughout the South Pacific, including American Samoa during 1979–1980, in which *Ae. polynesiensis* was implicated as a major vector [7, 8]. American Samoa has since experienced outbreaks of all four DENVs, and by 2010 more than 95% of adults were estimated to have been infected with at least one DENV [9]. Consistent with global trends in the emergence of other viruses transmitted by *Aedes aegypti*, outbreaks of chikungunya and Zika viruses occurred in American Samoa in 2014 and 2016, respectively [10, 11].

Due to trends in human movement and the limited flight range of peri-domestic *Ae. aegypti* mosquitoes, dengue cases typically cluster around households and areas where people congregate (e.g., schools, churches) [12, 13]; however, in American Samoa *Ae. polynesiensis* are more prevalent than *Ae. aegypti* [14, 15]. *Ae. polynesiensis* has long been known to be an effective DENV vector in the laboratory [16], although it has yet to be incriminated as a DENV vector in the wild [17]. Hence, the predominance of *Ae. polynesiensis* in American Samoa may result in differences in risk factors for DENV transmission compared to areas where *Ae. aegypti* is the dominant or sole vector. *Ae. polynesiensis* females tend to feed and rest outdoors more than *Ae. aegypti* females, which frequently enter houses and other buildings [18–21]. Both species have limited flight ranges [18, 22, 23], and both mainly utilize water-holding natural and artificial containers for larval development in American Samoa [14, 15]. In some settings, *Ae. aegypti* also develop in septic tanks or other underground water habitats [24–27].

During an outbreak of DENV-2 in American Samoa that started in 2016 [28], we sought to determine the incidence of DENV infection among household contacts of cases, identify risk factors associated with DENV infection, and characterize the distribution of known and suspected mosquito vectors of DENV. To do so, we conducted household-based cluster investigations that included collection of epidemiologic as well as entomologic data.

## Materials and methods

### Ethics statement

As this investigation was activity designed to identify, characterize, and control an immediate public health threat, following CDC Human Subjects Review it did not meet the definition of research and IRB review was not requested. Adults aged 18 years or older provided written consent for themselves and children aged <18 years for whom they were responsible. All minors provided assent if able. Consent, assent, and questionnaires were conducted in Samoan or English. Data were not anonymized prior to analysis, but were deidentified.

### Setting

American Samoa is located in the South Pacific Ocean, west of the Cook Islands and north of Tonga. It forms the eastern portion of the Samoan archipelago, while the nearby islands to the west comprise the independent country of Samoa. In 2010, the population of American Samoa was 55,519 [29], most (>95%) of which lived on the main island of Tutuila. Median age in American Samoa in 2010 was 22 years, and more than one-third of the population was aged <16 years. Most (92%) residents were Pacific Islander, and average household size was 5.6 people. Total land area of American Samoa is 76.8 square miles (199 square kilometers).

### Household-based cluster investigations

Household cluster investigations were conducted during September 25–October 3, 2017.

Investigation teams consisted of an interviewer, phlebotomist, and Environmental Health Service Officer. All cluster investigations were conducted on Tutuila, where all reported dengue cases resided. Patients reported to the American Samoa Department of Health (ASDOH) with suspected dengue were eligible to be contacted for this investigation if they had tested positive for DENV infection in the 30 days prior to the date of report to ASDOH by detection of: 1) DENV non-structural protein-1 (NS1) by rapid diagnostic test (SD BIOLINE Dengue Duo, Abbott Laboratories, Chicago, IL) performed in American Samoa; or 2) DENV nucleic acid by reverse transcription polymerase chain reaction (RT-PCR) [30] performed at the Hawaii State Laboratory in Honolulu, HI. Eligible case-patients or their parent or guardian were contacted by telephone to orient them to the investigation. If interested in participating, household visits were scheduled during which participation was offered to all residents of the case-patient's household. In addition, on the same day as the household visit to the case-patient's household, teams walked door-to-door to visit all households within a 50-meter radius of the case-patient's household, orient all available heads-of-household to the investigation, and offer participation to them and other household members. Households were not replaced or revisited if contact could not be made with the head-of-household or if they chose not to participate in the investigation. While index case-patients were invited to participate, cluster investigations were still conducted if case-patients were unavailable to be offered or declined participation.

Among participating households, adult heads-of-household were asked to complete a questionnaire regarding household-level characteristics including housing structure, mosquito control methods, presence of screens on windows and doors, air conditioning, income, and presence of a septic tank. An individual questionnaire was administered to all participating household members, which collected information on recent febrile illness, medical and travel history, mosquito avoidance behaviors, time spent at home, and education level. Parents or guardians answered questionnaires by proxy for participants aged <8 years. Educational materials about dengue, the need to seek medical care for febrile illness, and recommended

approaches to prevention were provided to each household regardless of participation in the investigation.

Blood specimens were collected from each participant and stored at 4° C until serum could be separated, after which serum was frozen at -70° C and transported to CDC Dengue Branch in San Juan, Puerto Rico. All specimens were tested by Trioplex RT-PCR to detect DENV, CHIKV, and ZIKV nucleic acid [30] as well as anti-DENV IgM using a commercially available diagnostic assay (DENV Detect IgM Capture ELISA, InBios International, Inc, Seattle, WA).

## Entomologic surveys

Among households that provided consent for mosquito sampling, a Prokopack aspirator [31] was used to collect mosquitoes inside the house and from around the outside of the house starting from the immediate periphery of the house and working outward, each for 15 minutes. The inside and outside of the houses were surveyed for potential water-holding containers and scored if the container held water and the presence/absence of mosquito larvae or pupae. Fewer containers were found indoors, which allowed identification of immature mosquitoes to sub-genus, whereas this was not logistically feasible for outdoor containers.

The outdoor inspections for containers and immature mosquitoes covered the entire household yard, usually clearly delimited by hedgerows, driveways, roads, streams, trees, rocks, etc. Aspirated adult mosquitoes were transported in a cooler to the laboratory where they were identified to species using the taxonomic keys of Ramalingam and Huang [32, 33]. *Ae. aegypti* and *Ae. polynesiensis* females were sorted into pools by species, house, and indoor/outdoor collection. Pools containing 1–18 females per vial were stored in RNAlater at -20° C until shipped to CDC Dengue Branch in Puerto Rico for testing by Trioplex RT-PCR [30].

## Definitions

"Index cases" were defined as the reported, laboratory-positive case-patients with dengue that initiated each cluster investigation. "Clusters" were comprised of all households within a 50-meter radius of the index patients' household and included vacant and occupied homes. "Participants" were household members who answered the individual questionnaire and provided a serum specimen. Participants of household-based cluster investigations were defined as being "laboratory-positive" if their serum specimen tested positive for DENV infection by either RT-PCR or anti-DENV IgM ELISA. Participants that tested negative by both assays were defined as being "laboratory-negative."

## Data analyses

All data cleaning and analyses were performed using SAS 9.4 (SAS Institute Inc., Cary, NC). To avoid sampling bias, two index case-patients that participated in cluster investigations were excluded from analyses to identify risk factors associated with DENV infection. Generalized estimating equation (GEE) analyses assuming an exchangeable correlation matrix (equal correlations among dependent observations) with a Poisson distribution (logarithm link) were used to model bivariate associations (prevalence ratios) among individual and household characteristics, entomological factors, and the outcomes of DENV infection. The GEE method accounts for correlations in data of participants from the same household and cluster that might otherwise bias variance estimates. Confidence intervals were constructed using robust estimates for standard errors. Fischer's Exact Test was used for small cell sizes. Due to small number of participants with DENV infection, a multivariate model of characteristics associated with DENV infection was not performed.

The attributable symptomatic infection rate was calculated by subtracting the percentage of participants without evidence of DENV infection that reported an acute febrile illness in the

past three months from the percentage of participants with evidence of DENV infection that reported an acute febrile illness in the past three months.

For entomologic data, descriptive statistics were calculated to characterize the frequency with which mosquito species were detected in and around homes and the various water-containers that were observed to be colonized.

### Molecular epidemiology

Specimens from dengue case-patients identified during the 2016–2018 outbreak were forwarded to CDC Dengue Branch for molecular phylogenetic analysis. The DENV-2 envelope glycoprotein (E) gene from 10 specimens amplified in tissue culture was sequenced using a Sanger bi-directional method described previously [34]. Briefly, primers specific for DENV-2 were used to amplify the E gene by RT-PCR. The amplification product was purified and sequenced with eight sequencing reactions on an ABI3500 Genetic Analyzer instrument (ThermoFisher). Genotyping analysis was performed by phylogenetic comparison of the 10 complete E gene sequences (1,485 basepairs) obtained in this study with 80 additional reference sequences retrieved from GenBank representing various DENV-2 genotypes. Sequences were aligned using MAFFT and a Bayesian phylogenetic tree was reconstructed using the BEAST v1.8.4 package [35, 36]. All sequences obtained in this investigation were published in GenBank (accession numbers MK244386–MK244395).

## Results

### Household cluster investigations

A total of 21 household-based cluster investigations were conducted around the residences of 21 index cases. Of 142 housing structures in all clusters, ten (7.0%) were vacant. Of the 132 occupied households, a head-of-household from 98 (74.2%) was offered participation in the investigation, of whom 97 (99.0%) accepted. Of 573 residents of all participating households, 252 (44.0%) were either not available or declined participation. Of the remaining 321 residents, 228 (71.0%) participants both completed the survey and provided a serum specimen, including two index case-patients that were removed from additional analyses. Among 226 participants that were included in the analysis, median age of participants was 35 years (range: <1–94), and more than half (56.2%) were female.

A total of seven (3.1%) participants tested laboratory-positive for DENV infection, of which two tested positive by RT-PCR and five by IgM ELISA. No participants tested positive for infection with CHIKV or ZIKV. Participants laboratory-positive for DENV infection resided in four (20.0%) of the 20 clusters and six (6.6%) of the 91 households.

Of 56 (24.8%) participants that reported a febrile illness in the past three months, seven (12.1%) were laboratory-positive for DENV infection. Febrile illness was reported among five (71.4%) participants that were laboratory-positive for DENV infection, and 51 (23.3%) that were laboratory-negative for DENV infection. The percentage of febrile illness attributable to DENV infection was 48.1%.

Following bivariate analysis, factors not significantly associated with DENV infection were age, sex, mosquito bite frequency, use of repellents, time spent at home, and use of a bed net (Table 1). The sole individual characteristic associated with DENV infection was reporting a febrile illness in the prior three months. Household characteristics associated with DENV infection included presence of a septic tank (Table 2). Although only four participants reported having burned citronella candles to attempt to control household mosquitoes, one tested positive for DENV infection, resulting in this uncommon behavior being significantly associated with DENV infection.

**Table 1. Association of individual characteristics with dengue virus infection among participants of household-based cluster investigations conducted in American Samoa, 2017.**

| Characteristic | Overall | | Laboratory-negative for DENV infection (n = 219) | | Laboratory-positive for DENV infection (n = 7) | | |
|---|---|---|---|---|---|---|---|
| | No. | (Column %) | Neg | (Row %) | Pos | (Row %) | PR (95% CI) |
| **Overall** | 226 | . . . | 219 | (96.9) | 7 | (3.1) | . . . |
| **Demographic** | | | | | | | |
| Female | 127 | (56.2) | 123 | (96.9) | 4 | (3.2) | 0.94 (0.18–4.75) |
| Male | 99 | (43.8) | 96 | (97.0) | 3 | (3.0) | Reference |
| Age in years, median (interquartile range) | 35 | (19, 49) | 36 | (19, 49) | 18 | (12, 50) | - |
| Age <18 years | 49 | (21.7) | 46 | (93.9) | 3 | (6.1) | 2.62 (0.77–8.89) |
| Age ≥18 years | 177 | (78.3) | 173 | (97.7) | 4 | (2.3) | Reference |
| **Education Level[1]** | | | | | | | |
| Grade 12 or above | 164 | (72.6) | 160 | (97.6) | 4 | (2.4) | 0.88 (0.27–2.87) |
| Less than grade 12 | 62 | (27.4) | 59 | (95.2) | 3 | (4.8) | Reference |
| **Medical** | | | | | | | |
| Febrile illness in past 3 months | 56 | (24.8) | 51 | (91.1) | 5 | (8.9) | 7.43 (1.85–29.83) |
| No febrile illness in past 3 months | 170 | (75.2) | 168 | (98.8) | 2 | (1.2) | Reference |
| **Behavioral** | | | | | | | |
| Mosquito repellent use in past month | 47 | (20.1) | 46 | (97.9) | 1 | (2.1) | 0.75 (0.12–4.68) |
| No mosquito repellent use in past month | 179 | (79.2) | 173 | (96.7) | 6 | (3.4) | Reference |
| Used bed net in past month | 26 | (11.5) | 25 | (96.2) | 1 | (3.9) | 1.05 (0.11–10.07) |
| Did not use bed net in past month | 200 | (88.5) | 194 | (97.0) | 6 | (3.0) | Reference |
| **Mosquito contact** | | | | | | | |
| Bitten by mosquitoes at least once per week | 140 | (62.0) | 135 | (96.4) | 5 | (3.6) | 1.80 (0.49–6.64) |
| Bitten by mosquitoes less than once per week | 86 | (38.1) | 84 | (97.7) | 2 | (2.3) | Reference |
| Time of day when mosquitoes bite[2] | | | | | | | |
| Evenings | 74 | (32.7) | 70 | (94.6) | 4 | (5.4) | 3.60 (0.60–21.72) |
| Not in evenings | 152 | (67.3) | 149 | (98.0) | 3 | (2.0) | Reference |
| Mornings | 61 | (27.0) | 61 | (100) | 0 | (0) | - |
| Daytime | 66 | (29.2) | 65 | (98.5) | 1 | (1.5) | - |
| Night-time | 70 | (31.0) | 65 | (92.9) | 5 | (7.1) | - |
| Where bitten by mosquitoes[3] | | | | | | | |
| Home | 158 | (69.9) | 152 | (96.2) | 6 | (3.8) | 2.81 (0.37–21.71) |
| Not at home | 68 | (30.1) | 67 | (98.5) | 1 | (1.5) | Reference |
| Work/School | 14 | (6.2) | 14 | (100) | 0 | (0) | - |
| Others' homes inside community | 10 | (4.4) | 9 | (90.0) | 1 | (10.0) | - |
| Others' homes outside community | 5 | (2.2) | 5 | (100) | 0 | (0) | - |
| Elsewhere | 11 | (4.9) | 10 | (90.9) | 1 | (9.1) | - |
| Mosquitoes do not bite me | 13 | (5.8) | 13 | (100) | 0 | (0) | - |

Abbreviations: DENV = dengue virus; PR = prevalence ratio; CI = confidence interval

[1] Education level was missing for 2 participants who were DENV negative.

[2] Univariable models were performed for participants who reported being bitten in the evening versus those that did not report being bitten in the evenings.

[3] Univariable models were performed for participants who reported being bitten at home versus those that did not report being bitten at home.

## Entomologic findings

From 90 unique premises that were inspected, we identified 552 outdoor containers with potential to become aquatic habitats for immature mosquitoes (Fig 1). The most common (n = 128, 23%) were plastic containers including cups, lids, and bags. Among the categories of

**Table 2. Association of household characteristics with status of dengue virus infection among participants of household-based cluster investigations conducted in American Samoa, 2017.**

| Characteristic | Overall | | Laboratory-negative for DENV infection (n = 219) | | Laboratory-positive for DENV infection (n = 7) | | |
|---|---|---|---|---|---|---|---|
| | No. | (Column %) | Neg | (Row %) | Pos | (Row %) | PR (95% CI) |
| **Household information** | 226 | . . . | 219 | (96.9) | 7 | (3.1) | . . . |
| Number of household members; median (IQR) | 7 (6, 9) | | 7 | (6, 9) | 6 (5, 11) | | |
| **Type of house** | | | | | | | |
| Single story | 189 | (83.6) | 184 | (97.4) | 5 | (2.7) | Reference |
| Two story | 21 | (9.3) | 20 | (95.2) | 1 | (4.8) | 1.99 (0.22–17.64) |
| Apartment | 16 | (7.1) | 15 | (93.8) | 1 | (6.3) | 2.02 (0.34–11.87) |
| Annual household income[1] <$15,000 | 104 | (54.5) | 103 | (99.0) | 1 | (1.0) | Reference |
| Annual household income ≥$15,000 | 87 | (45.6) | 82 | (94.3) | 5 | (5.8) | 6.22 (0.76–50.99) |
| **Screens on doors and windows** | | | | | | | |
| No screens | 24 | (10.6) | 24 | (100) | 0 | (0) | – |
| Some screens | 33 | (14.6) | 33 | (100) | 0 | (0) | – |
| Screens on all | 169 | (74.8) | 162 | (95.9) | 7 | (4.1) | – |
| **Air conditioning** | | | | | | | |
| None | 115 | (50.9) | 111 | (96.5) | 4 | (3.5) | 1.12 (0.23–5.40) |
| Any AC | | | | | | | Reference |
| Only in bedroom at night | 96 | (42.5) | 93 | (96.9) | 3 | (3.1) | – |
| In all rooms | 15 | (6.6) | 15 | (100) | 0 | (0) | – |
| **Leaves doors and windows open** | | | | | | | |
| Never | 108 | (47.8) | 103 | (95.4) | 5 | (4.6) | 0.37 (0.07–1.95) |
| Sometimes (combined daytime/nighttime/always) | | | | | | | Reference |
| Daytime only | 53 | (23.5) | 52 | (98.1) | 1 | (1.9) | – |
| Nighttime only | 3 | (1.3) | 3 | (100) | 0 | (0) | – |
| Always | 62 | (27.4) | 61 | (98.4) | 1 | (1.6) | – |
| **Mosquito Control Activities** | | | | | | | |
| Uses mosquito coils | 82 | (36.3) | 77 | (93.9) | 5 | (6.1) | 4.22 (0.81–21.95) |
| Does not use mosquito coils | 144 | (63.7) | 142 | (98.6) | 2 | (1.4) | Reference |
| Burning coconut husks | 11 | (4.9) | 10 | (90.9) | 1 | (9.1) | 3.30 (0.50–21.61) |
| Not burning coconut husks | 215 | (95.1) | 209 | (97.2) | 6 | (2.8) | Reference |
| Use lemon grass | 1 | (0.4) | 0 | (0) | 1 | (100) | – |
| Not use lemon grass | 225 | (99.6) | 219 | (97.3) | 6 | (2.7) | – |
| Burn citronella candles | 4 | (1.8) | 3 | (75.0) | 1 | (25.0) | 10.90 (1.23–95.97) |
| Not burn citronella candles | 222 | (98.2) | 216 | (97.3) | 6 | (2.7) | Reference |
| Spray insecticide indoor | 153 | (67.7) | 149 | (97.4) | 4 | (2.6) | 0.73 (0.14–3.84) |
| Not spray insecticide indoor | 73 | (32.3) | 70 | (95.9) | 3 | (4.1) | Reference |
| **Water supply** | | | | | | | |
| Piped water supply (municipal) | 213 | (94.3) | 206 | (96.7) | 7 | (3.3) | – |
| Any other source | 13 | (5.8) | 13 | (100) | 0 | (0) | – |
| Piped water supply (from village) | 21 | (9.3) | 21 | (100) | 0 | (0) | – |
| Rainwater | 11 | (4.9) | 11 | (100) | 0 | (0) | – |
| Bottled water | 18 | (8.0) | 17 | (94.4) | 1 | (5.6) | – |
| Other | 1 | (0.4) | 1 | (100) | 0 | (0) | – |
| **Household stores water** | | | | | | | |
| No | 148 | (65.5) | 141 | (95.3) | 7 | (4.7) | – |

(*Continued*)

**Table 2.** (Continued)

| Characteristic | Overall | | Laboratory-negative for DENV infection (n = 219) | | Laboratory-positive for DENV infection (n = 7) | | |
|---|---|---|---|---|---|---|---|
| | No. | (Column %) | Neg | (Row %) | Pos | (Row %) | PR (95% CI) |
| Yes | 78 | (34.5) | 78 | (100) | 0 | (0) | – |
| Water covered | 69 | (89.6) | 69 | (100) | 0 | (0) | – |
| Water not covered | 8 | (10.4) | 8 | (100) | 0 | (0) | – |
| **Home has septic tank** | | | | | | | |
| Yes[2] | 95 | (46.3) | 88 | (92.6) | 7 | (7.4)[3] | – |
| No | 110 | (53.7) | 110 | (100) | 0 | (0) | – |
| **Participated in community-led cleanup of mosquito breeding sites[4]** | | | | | | | |
| Yes | 76 | (33.6) | 73 | (96.1) | 3 | (4.0) | Reference |
| No | 150 | (66.4) | 146 | (97.3) | 4 | (2.7) | 0.63 (0.13–3.03) |

[1] Income was not reported by 35 participants: N = 191 for this variable.

[2] 21 households did not know if their house has a septic tank: N = 205 for this variable.

[3] Using Fisher's Exact Test to account for small cell size, having a septic tank was significantly associated with being dengue positive (p = 0.004)

containers surveyed, tires, buckets, and coconuts were the most common, representing a combined 27% of identified containers and 32% of containers that were producing pupae.

We found water-holding containers inside 26 (31%) of the 83 houses inspected. The most common indoor container was used to catch water underneath faulty plumbing, which was found in 19 of the households, five of which contained larvae or pupae of *Stegomyia* species. Other indoor containers included those used to root plant cuttings, collect water from leaking roofs, ant guards under table legs, and miscellaneous containers. Of these, immature *Stegomyia* species were found in the containers used for plant rooting and ant guards.

A total of 588 male and female mosquitoes of all species were collected, of which 311 (53%) were collected indoors (Table 3). Of the 183 females of the two putative vector species, 90.3%

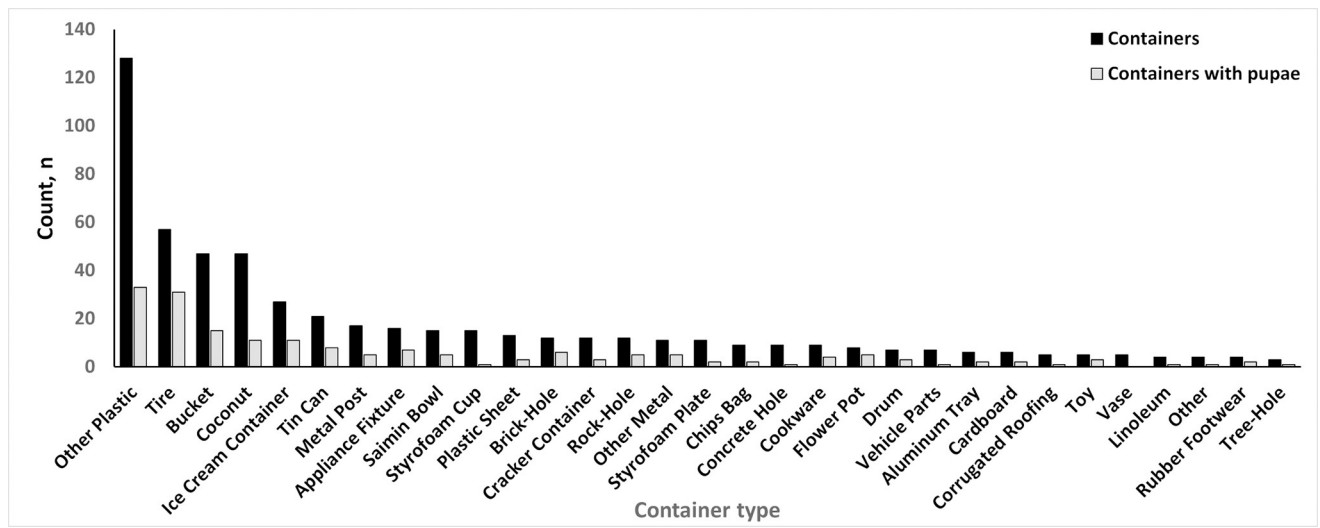

**Fig 1. Number of potential water-holding containers (black) and number of water-holding containers that contained pupae (gray) outside of homes in American Samoa, 2017.**

**Table 3. Adult mosquitoes collected from inside and outside households participating in dengue cluster investigations, American Samoa, September–October 2017.**

| Environment and species | Females | | | | | Males | | | |
|---|---|---|---|---|---|---|---|---|---|
| | Number collected | Percent of total | Range per home | Number (%) of homes with females | Mean females per home (standard error) | Number collected | Percent of total | Range for individual homes | Number (%) of homes with males |
| Indoors (N = 83) | | | | | | | | | |
| *Aedes aegypti* | 84 | 30.3 | 0–8 | 34 (40.9) | 1.0 (0.2) | 86 | 27.7 | 0–9 | 32 (38.6) |
| *Aedes polynesiensis* | 15 | 5.4 | 0–2 | 12 (14.5) | 0.2 (0.05) | 4 | 1.3 | 0–2 | 3 (3.6) |
| *Aedes upolensis* | 0 | 0.0 | 0 | 0 (0.0) | — | 0 | 0.0 | 0 | 0 (0.0) |
| *Aedes vexans* | 1 | 0.4 | 0–1 | 1 (1.2) | 0.1 (0.01) | 1 | 0.3 | 0–1 | 1 (1.2) |
| *Aedes (Finlaya) spp.** | 1 | 0.4 | 0–1 | 1 (1.2) | NC | 0 | 0.0 | 0 | 0 (0.0) |
| *Culex quinquefasciatus* | 64 | 23.0 | 0–8 | 15 (18.1) | 0.7 (0.21) | 55 | 17.7 | 0–15 | 13 (15.7) |
| Outdoors (N = 89) | | | | | | | | | |
| *Aedes aegypti* | 9 | 3.2 | 0–3 | 5 (5.6) | 0.1 (0.05) | 28 | 9.0 | 0–9 | 12 (13.5) |
| *Aedes polynesiensis* | 75 | 27.0 | 0–18 | 34 (38.2) | 0.8 (0.23) | 51 | 16.5 | 0–7 | 22 (24.7) |
| *Aedes upolensis* | 1 | 0.4 | 0–1 | 1 (1.1) | 0.01 (0.01) | 0 | 0.0 | 0 | 0 (0.0) |
| *Aedes vexans* | 5 | 1.8 | 0–3 | 3 (3.4) | 0.06 (0.04) | 55 | 17.7 | 0–31 | 4 (4.5) |
| *Aedes (Finlaya) spp.** | 4 | 1.4 | 0–1 | 4 (4.5) | NC | 1 | 0.3 | 0–1 | 1 (1.1) |
| *Culex quinquefasciatus* | 19 | 6.8 | 0–3 | 14 (15.7) | 0.2 (0.06) | 29 | 9.4 | 0–6 | 17 (19.1) |
| Total | 278 | 100 | | | | 310 | 100 | | |

Abbreviations: NC = not calculated

*As *Aedes oceanicus*, *Ae. samoanus*, and *Ae. tutuilae* are difficult to distinguish and were seldom found, these species were grouped into a single category

(n = 84) of *Ae. aegypti* and 16.7% (n = 15) of *Ae. polynesiensis* were collected indoors and 9.7% (n = 9) of *Ae. aegypti* and 83.3% (n = 75) of the *Ae. polynesiensis* were collected outdoors (Table 3).

None of the *Ae. aegypti* and *Ae. polynesiensis* pools were positive for detection of DENV, CHIKV, or ZIKV RNA by RT-PCR.

## Molecular epidemiology

Phylogenetic analysis of specimens from dengue case-patients demonstrated that the DENV-2 circulating in American Samoa during the 2016–2018 outbreak clustered with sequences of the Cosmopolitan genotype of DENV-2 and was closely related to recent sequences detected in the Solomon Islands and Papua New Guinea (Fig 2). Estimation of the time of the most recent common ancestor to the sequences obtained in this study suggests that the virus detected in 2017 in American Samoa had been circulating in the region for approximately 2.57 years (95% highest posterior density: 1.37–3.86 years).

## Discussion

In this investigation, we utilized household-based cluster investigations to identify factors associated with DENV infection in American Samoa and investigated the presence of DENV in both *Ae. aegypti* and *polynesiensis*. Although we were not able to incriminate *Ae. polynesiensis*

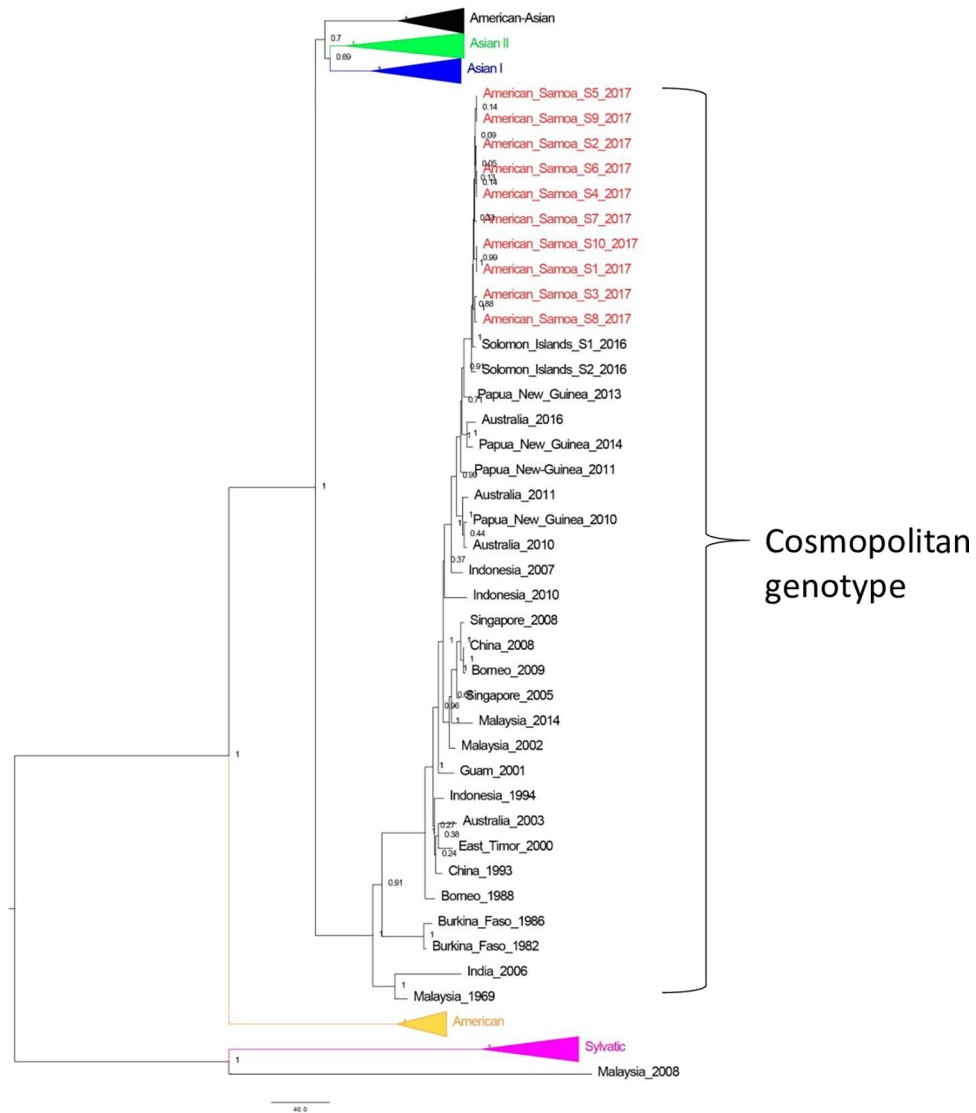

**Fig 2. Bayesian maximum clade credibility phylogenetic tree of DENV-2 detected during the 2016–2018 dengue outbreak in American Samoa.** N = 91 E gene sequences (1,485 basepairs). Internal nodes supported by posterior probability. The red text indicates DENV-2 detected in specimens collected from American Samoa in 2017 (GenBank accession number MK244386–95.1).

as a DENV vector in urbanized areas, recent febrile illness and presence of a household septic tank were significantly associated with DENV infection. These findings are consistent with the expected clinical manifestations of dengue and findings in other jurisdictions where abundance of *Ae. aegypti* was associated with the presence of septic tanks [1, 37, 38].

Significant association of household use of a septic tank with DENV infection is consistent with previous research from other jurisdictions in which septic tanks were identified as sites of prominent production of *Ae. aegypti* [24, 25]. Consequently, messaging to community members during the outbreak emphasized the importance of ensuring that septic tanks are sealed to the environment with netting or sealant that prevents mosquitoes from exiting. However, recent limited investigations demonstrated minimal production of *Aedes* spp. mosquitoes from septic tanks in one village of American Samoa, but substantial production of *Culex*

*quinquefasciatus* from some tanks. Moreover, as most septic tanks encountered were functioning improperly and may be present more often in communities with lower socioeconomic status, septic tanks in this jurisdiction may be a marker of communities with more traditional risk factors for DENV transmission [1]. Regardless, the relative contribution of septic tanks and above-ground water containers in American Samoa and elsewhere should be evaluated to further assess their potential as risk factors for DENV infection, including the impact of precluding them from serving as mosquito production sites.

Logistical constraints prevented us from quantifying and identifying pupae found in outdoor containers; however, findings from our outdoor immature survey agreed with what has been reported in Samoa and American Samoa [14, 39]. We identified fewer water storage drums than what was found in a similar investigation in 2007 [14], although the difference is likely attributable to variations in availability and quality of municipal water, and economic level of residents between villages sampled in the two studies as opposed to changes within villages between 2007 and 2017. Of note, this investigation identified containers collecting water from faulty plumbing inside homes as important production sites for *Stegomyia*, a finding that will be useful in future vector control initiatives.

Consistent with previous reports from American Samoa and elsewhere in the South Pacific [14, 15, 17], we detected *Ae. aegypti* mosquitoes predominantly indoors and *Ae. polynesiensis* mosquitoes predominantly outdoors. Aspirations have been reported to collect around 25% of the total indoor population of *Ae. aegypti* [40], suggesting the true abundance of *Stegomyia* in American Samoa is higher than what we recorded.

Although age was not significantly associated with DENV infection in this investigation, this may have been a result of the small number of DENV-infected participants identified. Among cases detected during this outbreak, incidence was highest among patients aged <20 years [28]. This observation is consistent with both the expected epidemiology of dengue, as well as a report of DENV-2 having circulated in the Pacific, including the neighboring island nation of Samoa, in 1997 [41]. Although we are unaware of documentation of circulation of DENV-2 in American Samoa in 1997 or soon thereafter, high attack rates in those years would be consistent with susceptibility of residents aged <20 years during the outbreak of DENV-2 during 2016–2018.

The molecular epidemiology of the DENV-2 circulating during the 2016–2018 dengue outbreak in American Samoa demonstrated that the closest relative was an isolate from Solomon Islands in 2016, and that divergence from the most recent known common ancestor occurred roughly two years prior. Both of these findings fit well with the index case-patient of the 2016–2018 outbreak being a fisherman who arrived in American Samoa in November 2016 one week after departing the Solomon Islands [28].

This investigation was subject to several limitations. First, a small (4%) proportion of household-based cluster investigation participants had evidence of acute or recent DENV infection, which decreased the power of the investigation to identify risk factors associated with infection. This small proportion may be attributable to the cluster investigations having been conducted relatively early in the course of the outbreak when only ~25% of all cases detected during the outbreak had been identified. Second, because neither the households nor the individuals that participated in this investigation were randomly selected, and because adults more often participated in investigations than children, the observed incidence of DENV infection should not be considered representative of all residents of American Samoa, neither at the time of the survey nor throughout the duration of the outbreak. Similarly, because this investigation was conducted during a public health response and was not designed to be representative of all residents of American Samoa, the findings should not be considered to be generalizable. Last, although questionnaires were administered in both English and Samoan,

only a standardized English version was available. As a result, interviewers translated questions during the interview, which may have resulted in variability in how questions were asked and consequent differences in interpretation of some questions by participants.

In conclusion, this report identifies septic tanks as an important risk factor associated with DENV infection in American Samoa. Further study is needed to determine whether septic tanks in American Samoa are significant sources of mosquito vectors of DENV or if the observed association between use of septic tanks and incidence of dengue is due to other factors. In many cases, interventions to block mosquito access to septic tanks could be implemented quickly, easily, and at relatively low cost. These findings will be of importance in the likely event of further dengue outbreaks in American Samoa and elsewhere in the Pacific. Similarly, although we confirmed that *Ae. aegypti* were more frequently present indoors whereas *Ae. polynesiensis* were predominantly encountered outdoors, future efforts should more systematically and longitudinally collect specimens to investigate the respective roles of *Ae. aegypti* and *Ae. polynesiensis* in transmitting DENV, CHIKV, and ZIKV.

## Supporting information

**S1 Data. Study data files.**
(XLSX)

**S2 Data. Study data files.**
(XLSX)

## Acknowledgments

We thank Rebecca Sciulli and Christian Whelen from Hawaii Department of Health State Laboratories Division for assistance with shipment of specimens to CDC Dengue Branch. We also thank Niela Leifi, Metotagivale Meredith, and Neil Gurr of the American Samoa Community College and Teejaye Maifea, Nu'ulua Sekai, Ioane Wesley, Joey Namuasua, Folasi Iosefo, John Mulu, Sonny Hleigh-Lord, and Fa'asisina Lake of the American Samoa Department of Health for assistance with collections of mosquitoes and recording of mosquito larvae habitats. We are grateful to all the families who participated in the surveys and allowed us to sample for mosquitoes in and around their houses.

Mary Mataia'a was unable to be contacted as of the time of the article's publication. The corresponding author vouches for her contributions to the work as reported in the article and is unaware of potential competing interests for her that would have impacted or been relevant to this work.

The findings and conclusions in this article are those of the authors and do not necessarily represent the official position of the U.S. Centers for Disease Control and Prevention or the U. S. Public Health Service.

## Author Contributions

**Conceptualization:** Tyler M. Sharp, A. John Tufa, Caitlin J. Cotter, Gilberto A. Santiago, Stephen H. Waterman, Jorge L. Muñoz-Jordán, Gabriela Paz-Bailey, Mark A. Schmaedick, Scott Anesi.

**Data curation:** Tyler M. Sharp.

**Formal analysis:** Matthew J. Lozier, Gilberto A. Santiago.

**Investigation:** Tyler M. Sharp, A. John Tufa, Caitlin J. Cotter, Gilberto A. Santiago, Stephanie S. Johnson, Mary Mataia'a, Mark A. Schmaedick, Scott Anesi.

**Methodology:** Tyler M. Sharp, Caitlin J. Cotter, Gilberto A. Santiago, Stephen H. Waterman, Jorge L. Muñoz-Jordán, Gabriela Paz-Bailey, Ryan R. Hemme, Mark A. Schmaedick, Scott Anesi.

**Project administration:** Tyler M. Sharp.

**Resources:** Tyler M. Sharp, Gabriela Paz-Bailey, Scott Anesi.

**Supervision:** Tyler M. Sharp, A. John Tufa, Stephen H. Waterman, Jorge L. Muñoz-Jordán, Mark A. Schmaedick, Scott Anesi.

**Validation:** Tyler M. Sharp.

**Visualization:** Tyler M. Sharp.

**Writing – original draft:** Tyler M. Sharp, Caitlin J. Cotter, Matthew J. Lozier, Gilberto A. Santiago, Mark A. Schmaedick.

**Writing – review & editing:** Tyler M. Sharp, Matthew J. Lozier, Gilberto A. Santiago, Stephen H. Waterman, Jorge L. Muñoz-Jordán, Gabriela Paz-Bailey, Ryan R. Hemme, Mark A. Schmaedick, Scott Anesi.

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
