## [Decision Letter · Decision Letter 0]

14 Nov 2022

PGPH-D-22-01116

Identification of Risk Factors and Mosquito Vectors associated with Dengue Virus Infection through Household-based Cluster Investigations in American Samoa

Dear Dr. Tyler M Sharp,

Thank you for submitting your manuscript to PLOS Global Public Health. After careful consideration, we feel that it has merit but does not fully meet PLOS Global Public Health’s publication criteria as it currently stands. Therefore, we invite you to submit a revised version of the manuscript that addresses the points raised during the review process.

We look forward to receiving your revised manuscript.

Kind regards,

Srinivasa Rao Mutheneni, PhD

Academic Editor

Journal Requirements:

2. Please send a completed 'Competing Interests' statement, including any COIs declared by your co-authors. If you have no competing interests to declare, please state "The authors have declared that no competing interests exist". Otherwise please declare all competing interests beginning with the statement "I have read the journal's policy and the authors of this manuscript have the following competing interests:"

3. Please amend your detailed Financial Disclosure statement. This is published with the article. It must therefore be completed in full sentences and contain the exact wording you wish to be published.

a State the initials, alongside each funding source, of each author to receive each grant.

4. Please provide separate figure files in .tif or .eps format only and ensure that all files are under our size limit of 10MB.

5. We do not publish any copyright or trademark symbols that usually accompany proprietary names, eg  ©, ®, ™  (e.g. next to drug or reagent names). Please remove all instances of trademark/copyright symbols throughout the text, including ™ on page 5 and ® on page 9.

6. Fig 1: please (a) provide a direct link to the base layer of the map (i.e., the country or region border shape) and ensure this is also included in the figure legend; and (b) provide a link to the terms of use / license information for the base layer image or shapefile. We cannot publish proprietary or copyrighted maps (e.g. Google Maps, Mapquest) and the terms of use for your map base layer must be compatible with our CC-BY 4.0 license. 

7. Please provide a complete Data Availability Statement in the submission form, ensuring you include all necessary access information or a reason for why you are unable to make your data freely accessible. If your research concerns only data provided within your submission, please write "All data are in the manuscript and/or supporting information files" as your Data Availability Statement.

Additional Editor Comments (if provided):

Reviewers' comments:

Reviewer's Responses to Questions

**Comments to the Author**

1. Does this manuscript meet PLOS Global Public Health’s publication criteria? Is the manuscript technically sound, and do the data support the conclusions? The manuscript must describe methodologically and ethically rigorous research with conclusions that are appropriately drawn based on the data presented.

Reviewer #1: Partly

Reviewer #2: Yes

Reviewer #3: Yes

Reviewer #4: Yes

2. Has the statistical analysis been performed appropriately and rigorously?

Reviewer #1: Yes

Reviewer #2: Yes

Reviewer #3: Yes

Reviewer #4: Yes

3. Have the authors made all data underlying the findings in their manuscript fully available (please refer to the Data Availability Statement at the start of the manuscript PDF file)?

Reviewer #1: Yes

Reviewer #2: Yes

Reviewer #3: Yes

Reviewer #4: Yes

4. Is the manuscript presented in an intelligible fashion and written in standard English?

Reviewer #1: No

Reviewer #2: Yes

Reviewer #3: Yes

Reviewer #4: Yes

5. Review Comments to the Author

Reviewer #1: The study entitled, "Identification of Risk Factors and Mosquito Vectors associated with Dengue Virus Infection through Household-based Cluster Investigations in American Samoa" is a simple study, which is based on questionnaire, and RT-PCR and anti-DENV IgM ELISA techniques. Although the study is a basic study but is well designed and effectively executed. However, the introduction section is very short and poorly framed. The rationale behind the study and the key questions of the research are not very clear. I think the authors need to reframe the introduction part. Moreover, in the methodology part, the authors need to mention a brief protocol (or the citation) for the identification of the mosquito species. Rest of the manuscript is ok.

Reviewer #2: Authors have described a household-based cluster investigation to identify population-specific risk factors associated with infection and performed entomologic surveillance to determine the relative abundance of Ae. aegypti and Ae. polynesiensis. The manuscript is well written and worth of publication at PLOS GPH, after minor revision.

It’s ok to use either mosquitos or mosquitoes, but please standardize throughout the manuscript. Additionally, some typing errors when writing Aedes species. Please review all manuscript carefully.

Author Summary.

Line 56 – Please add a space “some Aedes species…”

Line 62- Please correct. In the abstract and results section authors mentioned seven DENV-positive cases (ELISA = 7 and RT-PCR=2), but here says nine.

Materials and methods

Authors should add a space between number and the IU (celsius, min, etc.)

Results

I really recommend authors to provide a map showing the positive cases. Is there any proximity between locations? Any patterns?

Page 13, Lines 231-2 – Authors stated, “The percentage of febrile illness attributable to DENV infection was 48.1%.”

What are others febrile illness reported in the Island? It will be important to discuss more this finding.

Reviewer #3: In this manuscript the authors present the results of a household-based cluster investigation that was conducted during the 2017 outbreak of dengue fever in the American Samoa. The author’s findings are interesting in that they suggest that a significant number of containers of standing water were found that could be used for breeding, and indeed a considerable number were found to have pupae present in them. The linkage between the presence of a septic tank is also interesting, although as the author’s note, the sample size does limit the statistical power. Overall, the manuscript is well written, and the statistical analysis contained sufficient detail that I have only minor comments about the overall manuscript.

-- Minor Points --

While the authors provide some overview of the literature concerning the linkage between dengue and septic tanks in the discussion (starting at Line 271), it may be worth pointing to some of this literature in the introduction for those researchers that are still in the process of getting read-up on the dengue literature.

With regards to the above point, since the sample size is small enough, it might be worth noting the type of septic tank if that was recorded.

Minor typo in Table 2: lemon grass / limon grass

Reviewer #4: Congratulations, Although there were problems with the sample size, the paper is the result of a detailed analysis of the data and with potential for future projects.

Methods

Ethics statement: All research that involve human beings and collection of blood, tissues etc, needs an ethical approval. Do you have one?.

Household-based cluster investigations:

Please clarify if the patients reported with a suspected dengue (index cases), were tested again to dectect DENV, CHIKV and ZIKV.

Entomologic surveys:

The pupae or larvae you found in water-holding containers were collected to be identified at the laboratory? Please clarify

Results

You worked with 21 index cases? And then removed 2? Please clarify this.

You said in the methodology that you tested the patients against DENV, CHIKV and ZIKV, but I only see the results for DENV, do you have the results of infected people by CHIKV and ZIKV?

Would be desirable to describe a little bit the overall results of the questionnaires.

6. PLOS authors have the option to publish the peer review history of their article (what does this mean?). If published, this will include your full peer review and any attached files.

**Do you want your identity to be public for this peer review?** For information about this choice, including consent withdrawal, please see our Privacy Policy.

Reviewer #1: No

Reviewer #2: **Yes: **Rafael F C Vieira

Reviewer #3: No

Reviewer #4: No

---

## [Decision Letter · Decision Letter 1]

25 Jan 2023

Identification of Risk Factors and Mosquito Vectors associated with Dengue Virus Infection in American Samoa, 2017

PGPH-D-22-01116R1

Dear Tyler M Sharp,

We are pleased to inform you that your manuscript 'Identification of Risk Factors and Mosquito Vectors associated with Dengue Virus Infection in American Samoa, 2017' has been provisionally accepted for publication in PLOS Global Public Health.

Best regards,

Srinivasa Rao Mutheneni, PhD

Academic Editor

Reviewer Comments (if any, and for reference):

Reviewer's Responses to Questions

**Comments to the Author**

1. If the authors have adequately addressed your comments raised in a previous round of review and you feel that this manuscript is now acceptable for publication, you may indicate that here to bypass the “Comments to the Author” section, enter your conflict of interest statement in the “Confidential to Editor” section, and submit your "Accept" recommendation.

Reviewer #1: All comments have been addressed

Reviewer #3: All comments have been addressed

2. Does this manuscript meet PLOS Global Public Health’s publication criteria? Is the manuscript technically sound, and do the data support the conclusions? The manuscript must describe methodologically and ethically rigorous research with conclusions that are appropriately drawn based on the data presented.

Reviewer #1: Yes

Reviewer #3: Yes

3. Has the statistical analysis been performed appropriately and rigorously?

Reviewer #1: Yes

Reviewer #3: Yes

4. Have the authors made all data underlying the findings in their manuscript fully available (please refer to the Data Availability Statement at the start of the manuscript PDF file)?

Reviewer #1: Yes

Reviewer #3: Yes

5. Is the manuscript presented in an intelligible fashion and written in standard English?

Reviewer #1: Yes

Reviewer #3: Yes

6. Review Comments to the Author

Reviewer #1: The manuscript is Ok now after the revision.

Reviewer #3: With these revisions all of the concerns in the original review have been addressed - congratulations! The only thing that I noticed was that a typo was introduced on Line 171 with empty square brackets and the Ramalingam citation is missing.

7. PLOS authors have the option to publish the peer review history of their article (what does this mean?). If published, this will include your full peer review and any attached files.

**Do you want your identity to be public for this peer review?** For information about this choice, including consent withdrawal, please see our Privacy Policy.

Reviewer #1: No

Reviewer #3: No
